# A Review on Hydroponics and the Technologies Associated for Medium- and Small-Scale Operations

Roberto S. Velazquez-Gonzalez [1], Adrian L. Garcia-Garcia [1], Elsa Ventura-Zapata [2], Jose Dolores Oscar Barceinas-Sanchez [1] and Julio C. Sosa-Savedra [1,*]

1  CICATA Querétaro, Instituto Politécnico Nacional, Cerro Blanco 141, Colinas del Cimatario, Querétaro 76090, Querétaro, Mexico; rvelazquezg1900@alumno.ipn.mx (R.S.V.-G.); agarciag@ipn.mx (A.L.G.-G.); obarceinas@ipn.mx (J.D.O.B.-S.)

2  Centro de Desarrollo de Productos Bióticos, Instituto Politécnico Nacional, Calle Ceprobi No. 8, San Isidro, Yautepec 62739, Morelos, Mexico; eventura@ipn.mx

*  Correspondence: jcsosa@ipn.mx

**Abstract:** According to the Food and Agriculture Organization of the United Nations, the world population will reach nine billion people in 2050, of which 75% will live in urban settlements. One of the biggest challenges will be meeting the demand for food, as farmland is being lost to climate change, water scarcity, soil pollution, among other factors. In this context, hydroponics, an agricultural method that dispenses with soil, provides a viable alternative to address this problem. Although hydroponics has proven its effectiveness on a large scale, there are still challenges in implementing this technique on a small scale, specifically in urban and suburban settings. Also, in rural communities, where the availability of suitable technologies is scarce. Paradigms such as the Internet of Things and Industry 4.0, promote Precision Agriculture on a small scale, allowing the control of variables such as pH, electrical conductivity, temperature, among others, resulting in higher production and resource savings.

**Keywords:** hydroponics; Agriculture 4.0; small-scale production; sustainability

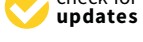



## 1. Introduction

The year 2050 harbingers a milestone in human history as, according to estimates from the United Nations for Food and Agriculture Organization (FAO), human population is expected to reach about 9 billion people [1]. To satisfy the demand for food, world production will have to raise about 70% from 2007 levels. Additionally, since the world is becoming increasingly urbanized, 75% of the world population is projected to live in urban settlements [2].

Currently, cities around the world occupy only 3% of the Earth's land, but account for 60–80% of energy consumption, 75% of carbon emissions [2], and 56% of the people [3]. The projections also indicate that world population growth will mainly occur in urban areas; that is, by 2050 urban population is expected to rise by 2.5 billion people, while total world population is projected to grow by somewhat less, 2.1 billion people [4]. Among the main factors fueling urban population growth are natural increase, migration from rural to urban locations, and reclassification: the geographic expansion of urban settlements at the expense of rural localities through annexation and transformation [4].

Both migration and reclassification pose agriculture to compete versus flourishing urban centers for soil, water, and human resources, forcing it not only to produce more food with fewer hands and less land, but also to fight climate change in all fronts: preserving habitats, protecting endangered species, and keeping biodiversity. Indeed, the burden on agriculture as the center piece of the food chain supply seems quite formidable. Nonetheless, open-field agriculture is still extensively used around the globe, despite the thousands of acres rendered useless for farming due to climate change, water scarcity, and soil pollution

by chemical pesticides and fertilizers [5]. As a result, driven by a growing demand of food in lockstep with population growth, agriculture needs to make decisive and radical moves towards efficiency and sustainability, using technology not simply for the sake of innovation, but to improve and address the real needs of consumers [6].

Since urbanization seems to be undeterred, it makes sense to include urban and peri-urban environments as part of the 2050 effort to feed the world with good quality, affordable, and sustainably produced food. In urban settlements, where space is limited and vegetative land uses are difficult to maintain, urban agriculture (UA), as is known, integrates into the urban economic and ecological system, the production of crop and livestock goods to provide products to the local population, including peri-urban agricultural areas around cities and towns [7]. The promise behind UA systems involves potential societal and environmental benefits for cities, such as enhanced food security and potential minimal environmental impact [8].

Those productive regions where arable land and water are becoming scarce, are turning to intensive high-yielding farming methods and technologies, like hydroponics, to meet the demand for healthy, affordable, and sustainable food. Europe and Asia Pacific regions are estimated to cultivate the most substantial amounts of tomatoes through hydroponics by 2028 [9]. When compared with traditional farming, hydroponics can produce higher yield by exploiting not only the horizontal surface area, but also the vertical space above it, effectively increasing the number of plants per unit area, and leaning towards vertical farming to meet daily consumer demands for nutritious fresh products in and around densely populated areas. Additionally, hydroponics makes it possible to harvest several crops throughout the year, without chaotic discharges of either pesticides or fertilizers to the environment, and using less land and water than traditional open-field agriculture. Indeed, by using smart greenhouses equipped with several technologies to control critical parameters for healthy plant physiology, hydroponics optimizes the use of water and chemicals to eliminate potentially hazardous waste and residuals [10].

Large-scale hydroponics facilities operate under controlled conditions of climate, lighting, and irrigation, rendered by numerous sensors, web platforms, software and mobile applications available nowadays. Due to such technological advancements, the hydroponics' market is expected to grow significantly from 2021 to 2028, at a compound annual growth rate (CAGR) of 20.7% from 2021 to 2028 [9]. Nonetheless, feeding human population by the milestone year of 2050 is not the only issue of concern; current global warming and generalized Earth's pollution are pressing environmental and socioeconomical concerns for which hydroponics may afford partial solutions to the shortcomings of traditional open-field agriculture: important contributions to the emissions of $CO_2$, and the loss of cultivable land due to outdated, unecological practices. The first step towards addressing 2050 human feeding needs is to embrace sustainability in all critical human activities, agriculture being one of them.

In contrast with the intrinsically unecological ways of nowadays open-field agriculture, hydroponics stands out as an appropriate, sustainable alternative for urban and peri-urban settlements, contributing to the sustainable development goal (SDG) number 11: sustainable cities and communities, of the United Nations' 2030 Agenda for Sustainable Development. The extended practice of hydroponics is essential for UA, and a game-changer in the supply chain of the food industry, the welfare of society, and the betterment of the environment; however, to reach its full potential, it needs to expand into the realm of small- and medium-scale production. It requires technologists to understand the basics of hydroponics to develop appropriate technologies, and producers to understand the benefits of applying new technologies to maximize cost-benefit at small- and medium-scale levels, to attend to either local consumers or self-consumption needs.

On the other hand, from an inchoate state, about 10,000 years ago, to present day, agriculture has transformed human societies and fueled human population growth [11]. Despite advances in plant physiology, agricultural methods have remained basically unchanged, adopting incremental technological advances in machinery and equipment, fertil-

izers, pesticides, and several other chemicals to boost productivity [12]. As a result, soil quality has been degraded to catastrophic levels, with minimum production, pressing for innovative, sustainable food systems. In this century, academy and industry have worked together to determine the main needs of agroindustry, and generate ad-hoc solutions to help agriculture move towards efficiency and sustainability, through a closer collaboration between relevant sciences.

The present review intends to close the gap between both fields, delving into the basics of hydroponics as a necessary step for the technologist to understand the needs of medium- and small-size operations, critical to achieve SDG 11. The work is organized as follows: In Part I, we describe the theoretical aspects related to the hydroponics technique. In Section 2, we review the historical contributions in hydroponics. Then, in Section 3, we examine the most common methods for growing crops using hydroponics. Section 4 reviews some advantages and disadvantages of implementing hydroponic systems. Section 5 includes a brief description of the substrates used in hydroponics must meet since these elements are the substitutes for agricultural soil. In Section 6, we review the ideal crops to develop in hydroponics. Sections 7 and 8 provide the reader with concepts related to nutrient solutions used in hydroponics and keeping them free from pathogens.

In Part II, we offer the reader an overview of the main technical aspects that contribute to the implementation of hydroponics in urban environments, with a small or medium-scale production approach. In Section 9, we provide an overview of how Industry 4.0 has revolutionized the world of agriculture, giving way to the concept of Agriculture 4.0. Section 10 reviews some technological approaches suitable for implementation in small and medium-scale agriculture. Finally, Section 11 shows a procedure for selecting the level of technology appropriate to the needs of farmers; in this section, we also review examples of autonomous systems applied to small-scale agriculture using hydroponics.

- **Part I. General Aspects of Hydroponics.**

## 2. History and Contributions from Plant Physiology

Hydroponics is a type of horticulture, a method that uses nutrient mineral solutions instead of tillage [13]. The oldest examples of hydroponics date as far back as the paintings on the walls of the Egyptian temple Deir El Bahari, more than four thousand years old [14]. During the VI century BCE, in Babylon, hydroponics was used to grow mostly ornate plants [15]. In pre-Columbian America, around the X and XI centuries CE, the Mexican Aztec culture developed the chinampa to grow crops on the shallow lake beds in the Valley of Mexico and believed to have been practiced throughout Mesoamerica [16]. Located in the outskirts of south Mexico City, Xochimilco's one thousand-year-old network of canals and artificial islands lionize the ingenuity of the Aztec people to make a sustainable habitat out of a wetland. In addition, it epitomizes the potential of UA as a sustainable food source and as an influential element in both social and environmental welfare. Consequently, UNESCO recognized this hydro-system as a World Heritage Site.

The development of hydroponics has gone in lockstep with our understanding of plant physiology. In 1600, Jean Baptiste Van Helmont, a Belgian scientist, conducted a series of experiments to demonstrate that plants can get some nutrients from water. Ninety-nine years later, British scientist John Woodward cultivated plants suspended over aqueous solutions and discovered that plants grew better in solutions enriched with fertilizer. Later, in 1800, French scientists De Saussure and Boussingault showed that plants require carbon, hydrogen, oxygen, and nitrogen to grow healthy. Then, in 1860, Sachs and Knop, in Germany, added phosphor, sulfur, potassium, calcium, and magnesium to the list of De Saussure and Boussingault, and grew plants in aqueous solutions containing salts of these elements [17].

Since then, scientific progress in the field of plant physiology has led to the knowledge that other elements such as manganese, molybdenum, chlorine, iron, zinc, copper, and boron, usually known as micro-nutrients, are needed for the healthy growth of plants, and

that the composition of the nutrient solution strongly correlates with the physiological response of plants in terms of size, color, and other crop features [18–20].

## 3. Hydroponic Cultivation Techniques

Hydroponics, unlike traditional farming, does not require soil to grow food. In this technique, plants are grown either on natural or man-made substrates, where the roots easily extract the nutrients from a prepared nutrient solution. There are different methods for growing food using hydroponics, and their application depends on the specific plant, local climate, and budget, among other factors. Most systems comprise a storage tank for the nutrient solution and an aerator, as illustrated in Figure 1.

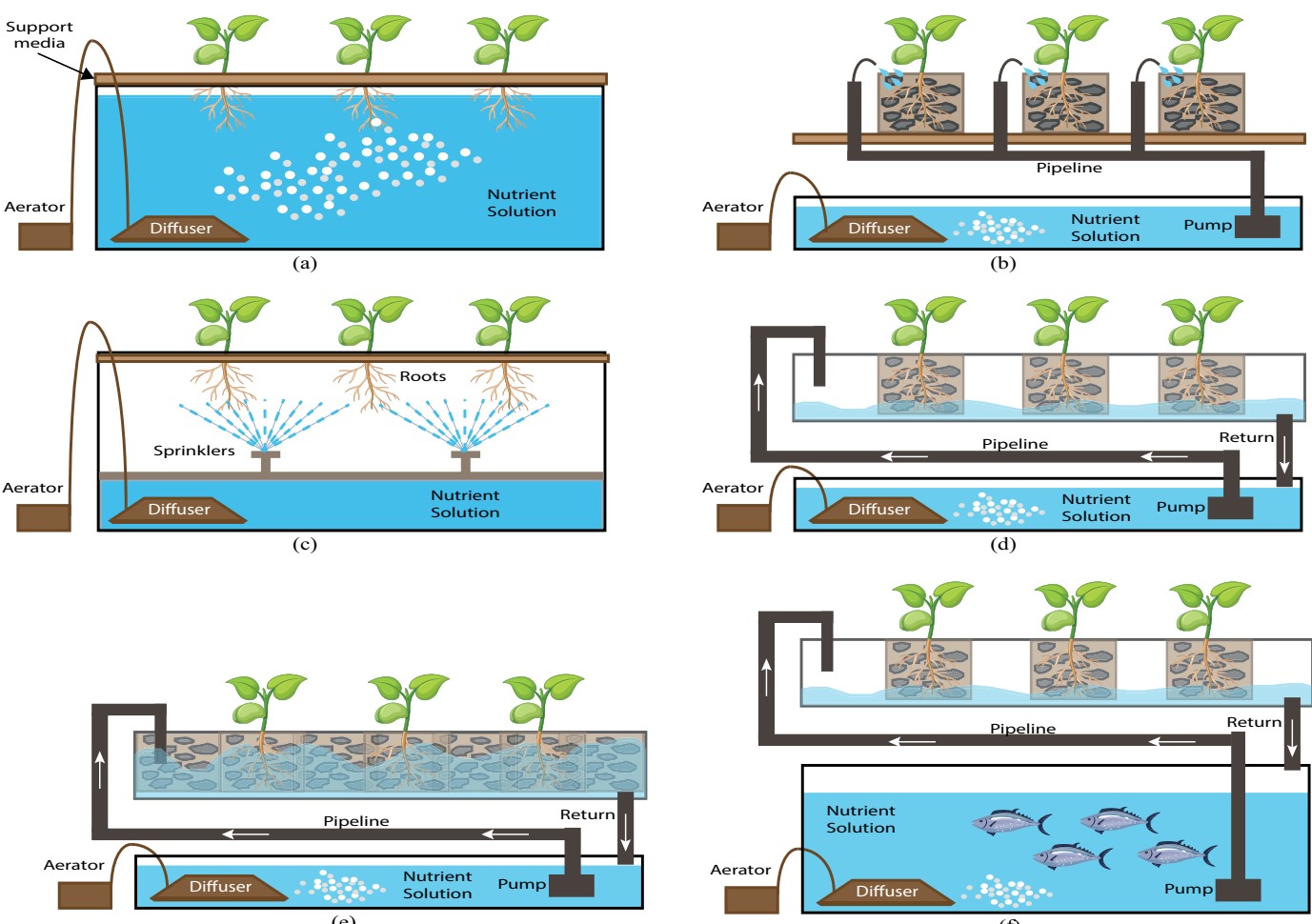

**Figure 1.** Different types of hydroponic systems. (**a**) Deep Water Culture. (**b**) Drip System. (**c**) Aeroponics. (**d**) Nutrien Film Technique (NFT). (**e**) Ebb and flow. (**f**) Aquaponics.

### 3.1. Floating Root System or Deep Water Culture (DWC)

In this system, the root of the plant is immersed in the nutrient solution, while the rest of it is supported above water level using polystyrene, cork bark or wood, among other materials (Figure 1a).

### 3.2. Drip Irrigation

This method is best suited for tomato and pepper-like crops. In this case, the nutrient solution is pumped directly to the roots of the plants with regulated flow. The solution is administered at predetermined time intervals and, for closed systems, the leftover solution is returned to the storage tank (Figure 1b).

### 3.3. Aeroponics

Tubers and roots are ideal to grow using aeroponics. In this configuration, the plants, with their roots hanging down in the air, get their nutrients from periodic spraying by a system of sprinkles. The main advantage of this technique is that it does not require an airing system as oxygen is carried along with the sprayed nutrient solution (Figure 1c).

### 3.4. Nutrient Film Technique

This method, also known as NFT, is like the floating root system, except that the plant roots are not completely submerged in the nourishing solution, but in a liquid stream flowing through a piping system. Although NFT requires smaller amounts of nutrient solution than the floating root system, it requires additional energy and components to operate. The excess solution returns to the storage tank by gravity and the flow of nutrient solution can be continuous or periodic (Figure 1d).

### 3.5. Ebb and Flow

Plants are placed in a tray, which is periodically filled with nutrient-rich water pumped from a reservoir below. The system uses gravity to return the water to the reservoir and reuse it (Figure 1e).

### 3.6. Aquaponics

This technique exploits the symbiosis of flora and fauna to achieve an efficient system in which fish feces afford the nutritional requirements of the plants. The absorption of nutrients by plants, combined with the microbial process of nitrification and denitrification, allows the recycling of water from the fish tank, forming a balanced micro-ecosystem (Figure 1f).

## 4. Advantages and Disadvantages

As illustrated in Figure 1, hydroponics offers many possibilities to grow food, but is not a panacea. Like any other agriculture technique, it poses opportunities and drawbacks that are worth reviewing to identify opportunity areas for the development of germane technologies.

Hydroponics, like any other agriculture technique, poses opportunities and drawbacks. These are worth reviewing to identify areas where technology could impact advantageously.

### 4.1. Advantages

- Ubiquity and space efficiency. Hydroponics makes it possible to grow food everywhere a controlled environment can be implemented. Indeed, even deep space exploration considers hydroponics as the main source of food for spaceship crews. Furthermore, depending on the type of plant, it is possible to devise vertical arrangements for increased produce output.
- Quantity and quality assurance. In traditional agriculture, crop rotation is necessary to preserve soil fertility; however, hydroponic crops can be repeated as many times as required, increasing the yield per cycle per crop. Also, since nourishment is administered accordingly with the physiological requirements of the plant, produce quality is assured. However, one must bear in mind that in highly automated greenhouses small changes in the operation conditions cause quick crop responses.
- Sustainability. Since the produce is not in contact with the soil and the nourishing solution is recycled, factors like water evaporation, seepage, or pollution are minimized and rinsing water is not needed. Additionally, by having a controlled environment, the optimal growth conditions and protection against plant plagues and diseases is assured, eliminating the need for chemicals and pesticides and saving on important natural resources like soil and water.
- Economics. In some routinary steps, operations in hydroponics are simpler than those required in traditional agriculture. In this sense, conventional practices require many effort-laden preparations before sowing, including the cost of heavy machinery and

specialized equipment, which eventually may come through a rental of the same. In other aspects, hydroponics may require more dedication, commonly needing a set of sensors and devices for a precise follow-up of the crop condition.

*4.2. Disadvantages*

- High initial cost. The initial investment in a hydroponic system is relatively high due to the cost of required raw materials and equipment for the operation.
- Highly trained labor. Large-scale hydroponic operations require personnel with deep knowledge of agriculture, plant physiology, chemistry, and sophisticated control and information systems.
- Environmental pollution. If the residual nutrient solution is not properly disposed of, the discharged solution, enriched with phosphorus and nitrates, can generate excessive growth of algae and other microorganisms in bodies of water and effluents, creating serious environmental problems.

**5. Substrates**

A substrate is the physical medium that supports plants by the stem and keep them under appropriate growing conditions by providing an aseptic environment with good oxygenation, and an adequate flow of the nutrient solution [21]. Table 1 is a compendium of typical materials used as substrates and describes their advantages and disadvantages. According to [15,22,23], some of the more relevant factors to look for when choosing a substrate are:

- Porosity. This property influences nutrient availability for the plant to perform metabolic processes like breathing, transpiration, and photosynthesis.
- Capillarity. Through capillarity, the substrate absorbs nutrients and distributes them to the plant's root.
- Oxygenation. The structure of the substrate must allow the intake of oxygen by the roots while these are in contact with the nutrient solution.
- Chemically inert. The substrate must consist of materials unable to react with the chemicals in the nutrient solution to avoid any alteration in its composition.
- Biologically inert. Because the nutrient solution circulates among a high root density of several plants, diseases can spread rapidly from one to another if corrective actions are not applied immediately. Therefore, the substrate must not favor any biological activity since micro-organisms may have a detrimental effect on crops, like diseases, malnutrition, and other consequences.

**Table 1.** Main substrates used in hydroponics.

| Material | Advantages | Disadvantages | Source |
|---|---|---|---|
| Sand | Economically viable, good porosity features, and provides good plant support | High density (around of 1500 kg/m$^3$), low retention of water, susceptible to salt accumulation | [22] |
| Perlite | Low density (around of 90 kg/m$^3$), biologically inert, neutral pH, highly available | Expensive, low water retention capacity | [23] |
| Vermiculite | Low density (around 80 kg/m$^3$), high nutrient holding ability, good water holding ability | Expensive, energy consuming product | [15] |

**Table 1.** *Cont.*

| Material | Advantages | Disadvantages | Source |
|---|---|---|---|
| Rockwool | Low density (around 80 kg/m$^3$), ease of handling, totally inert, sterile from pathogens, eases nutrition management in plants | Negative impacts on human health when is reused | [24,25] |
| Coconut coir | Low density (around 60 kg/m$^3$), good air content and water holding capacity, pH in ranges of 5–8 | High salt level content, energy consumption during transport | [26] |
| Peat/Peat moss | Inert, high water storage capacity, prevents leaching of nutrients | Negative environmental impacts such as loss of soil organic carbon, relatively expensive | [24] |
| Pumice | Cheap and long-lasting usage, chemically inert, low density | Particle size and hydraulic properties affect the growth and yield of crops | [27] |

*Sustainability Criteria for Choosing Substrates*

Some authors considered not only the chemical, biological or physiological aspects of substrates intended for hydroponic farming, but also their economic, social, and environmental viability. In his study, Gruda [25] criticizes the use of peat and rock wool as substrates, advocating for the use of materials that are more ecological, like organic waste and renewable raw materials, among others. Similarly, Rogers [28] considered compost as an alternative substrate, while Vinci and Rapa [23] analyzed life cycle impact assessment (LCIA) and life cycle cost (LCC) of rock wool, perlite, vermiculite, peat, coconut fiber, bark, and sand, to determine the social and economic impact of producing these materials. Additionally, they calculated the carbon footprint to measure their environmental impact.

## 6. Ideal Crops for Hydroponics

Hydroponics is appropriate to grow quite a variety of fruits and vegetables; however, these must meet certain criteria like root and fruit size, and harvesting time cycle, among others. Table 2 lists several crops suitable for hydroponics.

**Table 2.** Most suitable plants to grow by hydroponics [5].

| Type | Common Name | Scientific Name | Cultivation Technique |
|---|---|---|---|
| Bulb Vegetables | Garlic | Allium sativum | Drip irrigation |
| | Onion | Allium cepa | NFT, Drip irrigation |
| | Pore | Allium porrum | NFT, Drip irrigation |
| Leafy Vegetables | Lettuce | Lactuca sativa | NFT, DWC |
| | Cabbage | Brassica oleracea var. capitata | NFT, DWC |
| | Brussels sprouts | Brassica oleracea var. gemmifera | NFT, DWC |
| | Mustard | Brassica nigra | NFT, DWC |
| | Spinach | Spinacea oleracea | NFT, DWC |
| | Chard | Beta vulgaris var. cicla | NFT, DWC |
| | Water cress | Nasturtium officinale | NFT, DWC |
| | Celery | Apium graveolens | NFT, DWC |
| | Parsley | Petroselinum crispum | NFT, DWC |
| | Coriander | Coriandrum sativum | NFT, DWC, drip irrigation |
| | Purslane | Portulaca oleracea | NFT, DWC |

**Table 2.** *Cont.*

| Type | Common Name | Scientific Name | Cultivation Technique |
|---|---|---|---|
| Root Vegetables | Beetroot | Beta vulgaris | Drip irrigation, aeroponics |
| | Jicama | Pachyrrhizus erosus | Drip irrigation |
| | Turnip | Brassica rapa | NFT |
| | Radish | Raphanus sativus | Drip irrigation, aeroponics |
| | Yucca | Manihot esculenta | NFT, Drip irrigation |
| | Carrot | Daucus carota | Drip irrigation, aeroponics |
| Tuber Vegetables | Sweet potato | Ipomoea batatas | Drip irrigation |
| | Potato | Solanum tuberosum | Drip irrigation |
| Stem Vegetables | Swede | Brassica oleracea var. gongyloides | Drip irrigation |
| | Asparagus | Asparagus officinalis | NFT, DWC |
| Inflorescent Vegetables | Artichoke | Cynara scolymus | Drip irrigation |
| | Broccoli | Brassica oleracea var. Italica | Drip irrigation, NFT |
| | Cauliflower | Brassica oleracea var. botrytis | Drip irrigation, NFT |
| | Huauzontle | Chenopodium sp. | NFT, DWC |
| Fruit Vegetables | Zucchini | Cucurbita pepo | Drip irrigation, NFT |
| | Cucumber | Cucumis sativus | Drip irrigation, NFT |
| | Cantaloupe | Cucumis melo | Drip irrigation, NFT |
| | Watermelon | Citrullus vulgaris | Drip irrigation |
| | Green bean | Phaseolus vulgaris | Drip irrigation |
| | Squash | Sechium edule | Drip irrigation |
| | Chile | Capsicum annuum | Drip irrigation |
| | Eggplant | Solanum melongena | Drip irrigation |
| | Tomato | Solanum licopersicum | Drip irrigation, NFT |
| | Tomato | Physalis ixocarpa | Drip irrigation, NFT |
| Pulse Vegetables | Pea | Pisum sativum | Drip irrigation |
| | Bean | Vicia faba | Drip irrigation |
| | Sweet Corn | Zea mays | Drip irrigation |

## 7. Nutrient Solution

In hydroponics, all essential nutrients are provided to the plant via the nutrient solution, except for carbon, hydrogen, and oxygen, which are air borne. Inorganic fertilizers are used as nutrient sources, except for iron, which is added as a chelate to improve its availability. Most of the fertilizers used in hydroponics to prepare nutrient solutions are highly-soluble inorganic salts; however, some inorganic acids are also used [29]. Plant nutrition in hydroponics has been extensively studied and the nutrients involved have been grouped in three kinds: primary, secondary, and trace or micro-nutrients, as summarized in Table 3.

**Table 3.** Elements absorbed by plants [5].

| Nutrient | Symbol | Forms Absorbed |
|---|---|---|
| Nitrogen | N | $NO_3^{2-}, NH_4^+$ |
| Phosphorus | P | $PO_4^{3-}, HPO_4^{2-}, H_2PO_4^-$ |
| Potassium | K | $K^+$ |
| Calcium | Ca | $Ca^{2+}$ |
| Magnesium | Mg | $Mg^{2+}$ |
| Sulfur | S | $SO_4^{2-}$ |
| Iron | Fe | $Fe^{2+}, Fe^{3+}$ |
| Manganese | Mn | $Mn^{2+}$ |
| Zinc | Zn | $Zn^{2+}$ |
| Copper | Cu | $Cu^{2+}$ |
| Molybdenum | Mo | $MoO_4^{2-}$ |
| Boron | B | $BO_3^{2-}, B_4O_7^{2-}$ |

The chemical composition of the nutrient solution depends on specific crop and plant development stage. Some soluble fertilizers used in hydroponics are ammonium nitrate ($NH_4NO_3$), calcium nitrate ($5[Ca(NO_3)_2 \cdot 2H_2O]NH_4NO_3$), phosphoric acid ($H_3PO_4$), nitric acid ($HNO_3$), etc [29]. Although these formulations are commercially available in liquid or solid presentation, it is also possible to prepare from scratch the mixture of salts, minerals and fertilizer. However, it is important to emphasize that, depending on their growth stage, plants require different formulations; thus, during the vegetative state, the plant grows foliage until ready to flower or root ripening, at which point a nutrient solution rich in phosphorous is required by the plant to build strong roots. Finally, during fruit ripening, the plant requires nutrient solutions low in N and high in K concentrations.

Commercially available solutions codify the macro-nutrient contents as a three-digit sequence, according to the N-P-K concentration expressed in weight percent. For instance, an 8-15-36 formulation, ideal for tomato crops, contains 8% N, 15% P, and 36% K. For lettuce crops, an 8-15-16 solution is recommended. It is also possible to add organic nutrients like compost: a mixture of vegetable waste, urine, manure, and dead animal parts, by leaching the solution before adding it to the hydroponic system. This type of tea serves as a possible alternative to the inorganic fertilizers commonly used in hydroponics. However, adding such materials to the formulation may contaminate the system with undesired parasites or bacteria, which is why it must be carefully analyzed before introducing it into the hydroponic system.

The lifetime of the solution is of the utmost importance and will depend on timely adjustments made to the pH, electrical conductivity, and water level. To exclude changes in the nutrient solution, the volume level in the storage tank must remain constant, replenishing the water absorbed by the plants and lost by evapotranspiration; otherwise, the concentration of the salts will change, affecting the healthy growth of the plants. Bosques [30] recommends changing the solution in the tank every 2 to 3 weeks, depending on crop, thoroughly cleaning and disinfecting the tank.

### 7.1. pH in Hydroponics Nutrient Solutions

An important chemical property of a nutrient solution is its pH, a scale of 1 to 14 used to specify acidity or alkalinity of a solution. At room temperature, water is neither basic nor acidic, hence it has been assigned a pH of 7. Solutions with pH higher than 7 are basic, and acidic otherwise. Most authors agree that the nutrient solution must have pH between 5 and 7 [31], since it is in such interval that nutrients remain soluble. However, if pH > 7, the solubility of Fe and $H_2PO_4^-$ decreases, giving rise to Ca and Mg precipitates, among other chemical reactions between nutrient solution components, hindering the absorption of iron, boron, copper, zinc, or manganese. On the other hand, if pH is below 5, the adsorption of nitrogen, phosphorus, potassium, calcium, magnesium, and molybdenum is inhibited. Table 4 shows the optimum pH range to grow some of the most popular vegetables. In some cases, the delivery of some micro-nutrients, like manganese, may result in hazardous pollution [30]. Figure 2 shows the availability of some nutrients as a function of pH.

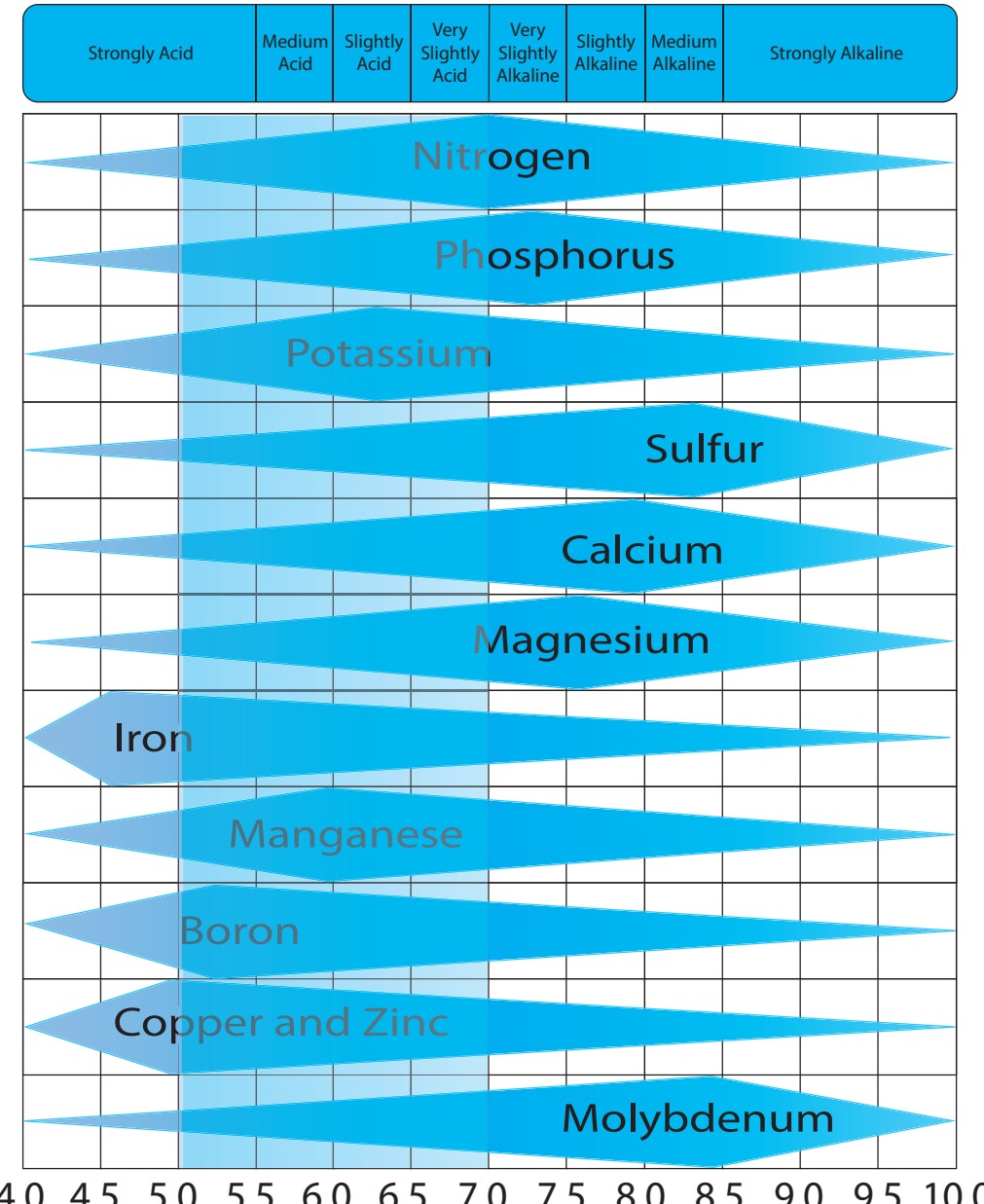

**Figure 2.** Effect of pH on the availability of nutrients. Out of the 5–7 pH interval, nutrient availability gets compromised by chemical reactions between the components of the nutrient solution [32].

**Table 4.** Optimum range of electrical conductivity (EC) and pH for some popular hydroponic crops [33].

| Crops | pH | EC (mS/cm) |
|---|---|---|
| Asparagus | 6–6.8 | 1.4–1.8 |
| African Violet | 6–7 | 1.2–1.5 |
| Basil | 5.5–6 | 1–1.6 |

**Table 4.** *Cont.*

| Crops | pH | EC (mS/cm) |
| --- | --- | --- |
| Bean | 6 | 2–4 |
| Banana | 5.5–6.5 | 1.8–2.2 |
| Broccoli | 6–6.8 | 2.8–3.5 |
| Cabbage | 6.5–7 | 2.5–3 |
| Celery | 6.5 | 1.8–2.4 |
| Carnation | 6 | 2–3.5 |
| Courgettes | 6 | 1.8–2.4 |
| Cucumber | 5–5.5 | 1.7–2 |
| Eggplant | 6 | 2.5–3.5 |
| Ficus | 5.5–6 | 1.6–2.4 |
| Leek | 6.5–7 | 1.4–1.8 |
| Lettuce | 6–7 | 1.2–1.8 |
| Marrow | 6 | 1.8–2.4 |
| Okra | 6.5 | 2–2.4 |
| Pak Choi | 7 | 1.5–2 |
| Peppers | 5.5–6 | 0.8–1.8 |
| Parsley | 6–6.5 | 1.8–2.2 |
| Rhubarb | 5.5–6 | 1.6–2 |
| Rose | 5.5–6 | 1.5–2.5 |
| Spinach | 6–7 | 1.8–2.3 |
| Strawberry | 6 | 1.8–2.2 |
| Sage | 5.5–6.5 | 1–1.6 |
| Tomato | 6–6.5 | 2–4 |

*7.2. Electrical Conductivity*

Electrical conductivity, EC, is an estimation of the total concentration of ions in a solution. In this case, low values of EC indicate a scarcity of nutrients in the form of ions; on the other hand, too-high values may lead to salt stress in the plant [26]; thus, EC should be kept within a target range because it significantly affects growth and crop quality [34]. Additionally, this parameter does not provide specific information regarding the concentration of each element in the nutrient solution; hence, after measuring EC, it is essential to add fertilizers in concentration amounts that the plants can absorb. Table 4 indicates adequate ranges of EC and pH for some popular crops.

**8. Sterilization of Nutrient Solutions**

An aseptic environment is of the utmost importance in hydroponic systems, to efficiently produce good quality products; however, it is hard to keep sterile the zone around the plant's roots [35]. The most obvious symptom of a diseased plant is leaf withering, caused by Fusarium and Verticillium fungi. Other species of parasites like Pythium and Phytophthora also represent menaces to the plant's roots. Unfortunately, there is no safe enough fungicide to use in hydroponics without compromising consumer health [36].

From a sustainability perspective, it is important to recirculate the nutrient solution to minimize water consumption and residuals to dispose of; however, it is not always possible to implement systems that balance the consumption of natural resources, energy, and financial costs. Although there are several techniques to prevent infections in hydroponic solutions, summarized in Table 5, given their specific advantages and disadvantages, a combination of them could be a better approach to the sepsis problem of the nutrient solution.

**Table 5.** Sterilization methods for nutrient solutions: pros and cons [37,38].

| Method | Advantages | Disadvantages |
|---|---|---|
| Filtration | | |
| -Sand Filters | Low cost<br>Easy to operate | High space requirements<br>Effectiveness varies with pathogen |
| -Membrane | Highly effective | Frequent clogging and seeping<br>High initial investment<br>Expensive maintenance |
| Heat treatment | | |
| -Pasteurization | Highly effective<br>Precipitates are not generated | High capital costs<br>High maintenance costs |
| Radiation | | |
| -UV Radiation | High efficiency (without turbidity)<br>Low space requirement | Low efficiency in the presence of turbidity<br>Precipitation of Mn and Fe<br>Relatively expensive equipment |
| Chemical treatment | | |
| -Ozone | Highly effective | High capital costs<br>High maintenance costs<br>Efficiency drops with high organic matter |
| -Hydrogen peroxide | Useful for cleaning irrigation systems | Interaction with some micronutrients<br>Harmful for plant roots when the dose is greater than 0.05% |
| -Chlorine | Low cost technique | Its effectiveness depends on many factors: temperature, pH, organic load, ammonium content, etc.<br>Toxic residues can be generated due to the interaction with organic and inorganic elements of the nutrient solution |

*Impact of Residual Nutrient Solutions on the Environment*

A by-product of either open field or protected cultivation is the residual chemical solution, rich in plant nutrients that may harm the environmental if not properly handled. Leftover nutrient solutions from hydroponic farming contains large amounts of nitrates (200–300 mg $NO_3^- \cdot L^{-1}$) and phosphates (30–100 mg $4PO_4^- \cdot L^{-1}$) that promote the growth of algae in [39].

In open field cultivation, rain may carry an excess of nutrients to rivers, eventually reaching lakes and oceans. For instance, recent studies have determined that the so-called dead zone in the Gulf of Mexico (Figure 3), occurring annually, is due to the discharges from the Mississippi River that, during its course across the United States, collects water enriched with nutrients produced from human activity, and fertilizers. Once the polluted waters reach the ocean, the excess of nutrients encourages the inordinate growth of algae that, after dying, sink and decay, consuming high oxygen levels. Without enough oxygen, marine life is under stress, with a long-term impact on species that cannot leave the area [40]. On the other hand, a sustainable way of managing residual-nutrient solutions in hydroponics could be by implementing closed systems that recover the water at the end of the cycle through treatment and sterilization, as schematized in Figure 4. Montesano et al. [41] have proposed the utilization of dielectric moisture sensors in soilless agriculture to improve the efficiency of water usage. Their results indicate that a network of wireless sensors for real-time monitoring of substrate humidity, combined with precise information of the effect of water availability levels on basil, may help optimize water consumption in such crops.

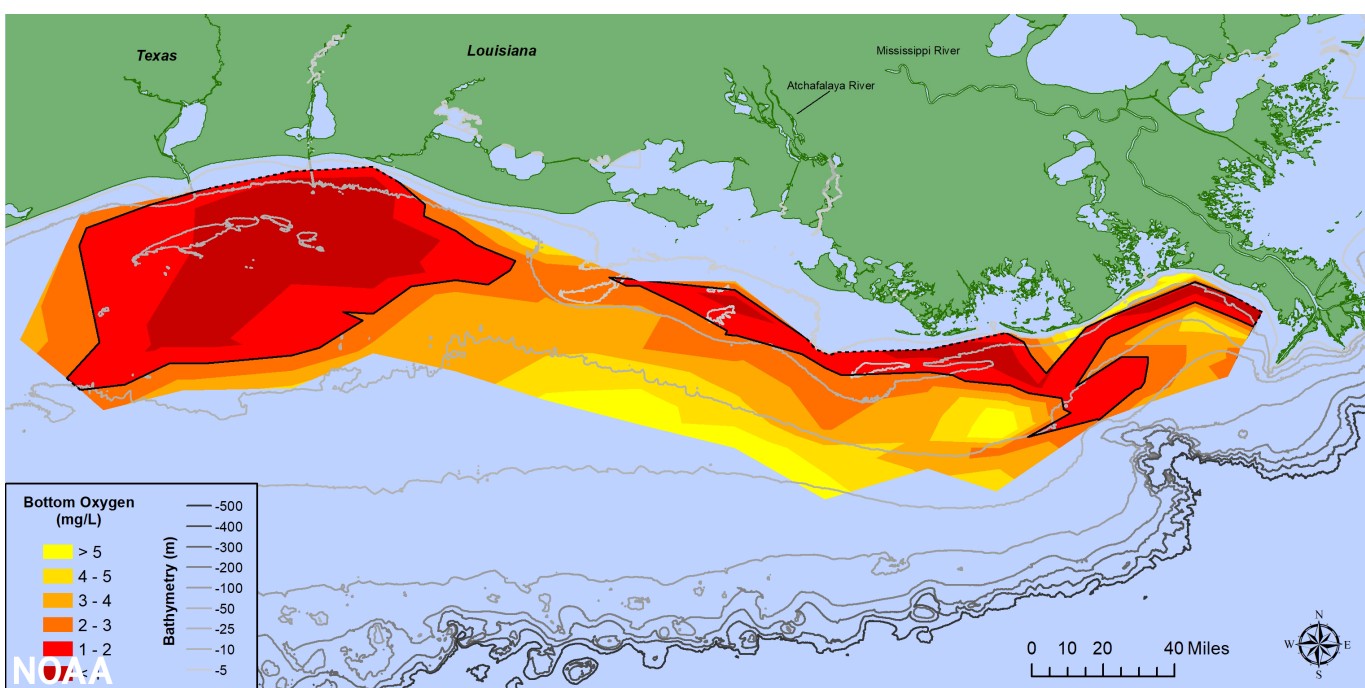

**Figure 3.** Map of measured Gulf hypoxia zone, 25–31 July 2021. (LUMCON/NOAA).

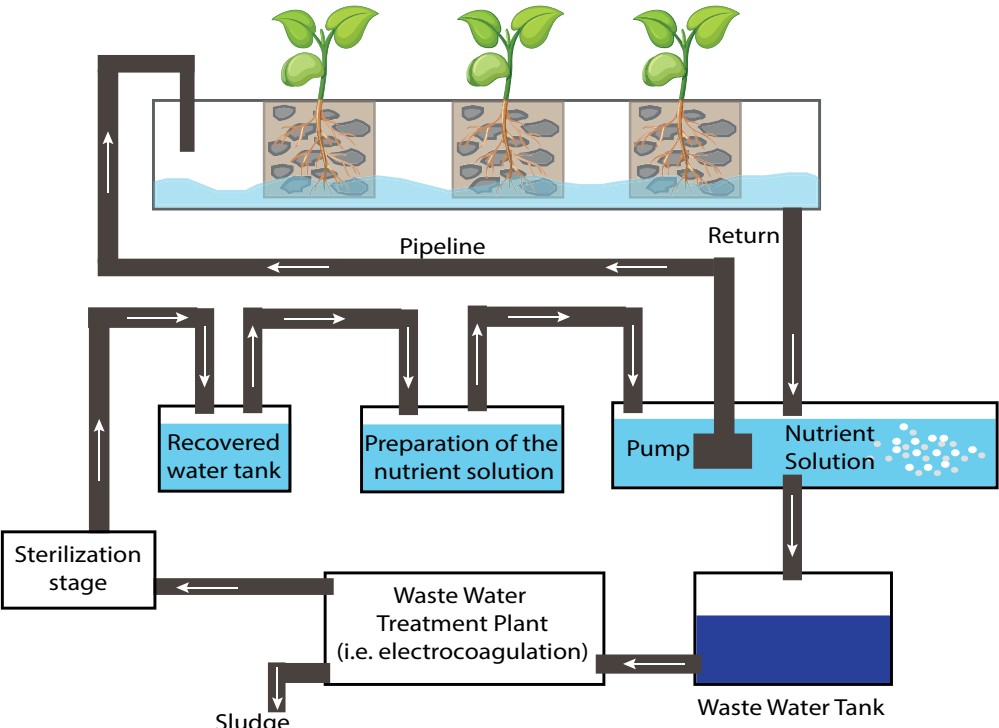

**Figure 4.** Implementation of a treatment system for water reuse in hydroponics systems.

- **Part II. Hydroponics and Technology.**

As stated in the introduction, several factors like human population growth and farmland degradation drive the global demand for more grains and food, pushing for higher yield and efficiency farming techniques, such as precision and urban agriculture. The restriction of space for cultivation in urban environments has called for efficient forms of cultivation. Based on hydroponics, technological innovation, and state of the art scientific knowledge, methods like vertical and indoor cultivation have extended rapidly, as attested by the market size and growth rate (CAGR 11.3%) of hydroponics, valued at USD 9.5 billion

in 2020, and projected to reach USD 17.9 billion by 2026 , thanks to the growing acceptance of indoor farming, and innovative technological advances.

## 9. Agriculture 4.0

Modern farms and agricultural operations are changing rapidly; technological innovation in electronics have led to the adoption of machines, temperature and moisture sensors, aerial imaging, and GPS, to revolutionize farming. The notion of industry 4.0 in the field of agriculture was born in 2017, where concepts such as artificial intelligence (AI), Internet of Things (IoT) and Big Data (BD) were integrated to autonomous food production systems for precision irrigation, pest control, plant disease identification, and production management [42]. Agriculture 4.0, as this revolution is known, intends to seamlessly join agricultural practices with state-of-the-art technology, including sensors, devices, machines, and information technology. Intricate technologies such as robots, temperature and moisture sensors, aerial and satellite imaging, and GPS technology are becoming increasingly used to enhance the entire food value chain and cause businesses to be more profitable, efficient, safer, and ecological [6].

Hydroponics fits perfectly within the frame of Agriculture 4.0, as large companies are increasing the use of breakthroughs in indoor vertical farming, artificial intelligence, and Plant Biology to grow an extensive line of products [43]. With the support of advanced and disruptive technologies and sound scientific knowledge to warrant high yield, we can say, with certainty, that hydroponics has clinched a center place in future food production systems. The challenge at hand is to bring these technological advances down to the medium- and small-scale operations found in urban and peri-urban settlements, where hydroponics may contribute significantly to achieve SDG 11: sustainability and resiliency of urban communities.

The assimilation of Agriculture 4.0 philosophy is by no means automatic, many hurdles need to be removed before terms like ecoagriculture [44], agrophotovoltaics [45], and precision agriculture [46] become part of the culture and practices of farmers around the world. To achieve it, agriculture and technology will have to come to common grounds for both producers and technologists. The former, to understand the best uses of technology and demand innovations that address the real needs of the food supply and value chains; the latter, to fulfill the expectations of producers with highly improved products, services, and processes to support sustainable and efficient food production in urban and peri-urban settlements.

## 10. Suitable Technology for Small and Medium-Scale Food Production Using Hydroponic

Indoor production has led to the development of small businesses, thus stimulating the growth of local economies. In Denmark, Avgoustaki and Xydis [47] compared the profitability of a business based on indoor urban vertical farming (IUVF) with that of a greenhouse. The results show that at small- to medium-scale, the profitability of IUVF is better than that of the greenhouse, due to the maintenance cost of the greenhouse. The authors conclude that product prices must be raised to 10 euros per kilogram to register an economic profit, without the need for government subsidies.

Indoor farming uses different types of equipment, including heating, ventilation, and air conditioning systems (HVAC), fans, irrigation systems, control systems, rails, and lights, which constitute a key cost factor to consider when setting up a hydroponics farm. The huge initial cost of the system, estimated at USD$ 110,000 for a 46.5 m$^2$ not fully automated farm [48], is a major hindrance in the adoption of hydroponics as a farming method. Hence, the importance of developing new and better products and services to support UA based on hydroponics. For a significant contribution to reaching SDG 11, such technologies must be scalable to fulfill the needs of growers, not only for large-scale operations but also for medium- and small-scale, considering the limitations of space available for farming. Buehler and Junge [49] have shown that the practice of growing

food in urban environments has grown significantly since 1988 in Europe, Asia, and North America, where the dedicated surface is more than 150,000 m$^2$. The practice of hydroponics is key for the emplacement of UA, and it may contribute significantly to reaching SDG 11; however, it requires the development and adoption of appropriate technologies.

Some technologies appropriate for indoor farming are presented by Gnauer et al. [50], who propose a frame work that integrates heterogeneous devices on different computing layers to monitor and optimize the production process. Additionally, they developed a robot for microgreens production called AgroRobot, which makes use of aeroponics. The AgroRobot is controlled by an Arduino Nano8 to control irrigation and lighting; furthermore, it has a touch-screen graphic interface designed with Nexion. This work also makes use of 3D printing technology to manufacture some accessories, such as culture trays.

Applications of Agriculture 4.0 into the realm of hydroponics are well on their way. For instance, Mycodo Environmental Regulation System is an open-source project in which a configurable graphical interface for indoor hydroponic production of leafy species was developed [51]. Mycodo runs in a Linux environment, specifically in the kernel of a Raspberry Pi. The technological contribution of this system is its scalability, i.e., in the interface designed by the authors, it is possible to add calibration routines for sensors, and control algorithms for variables such as pH, EC, humidity, among others. In addition, they used additive manufacturing for some accessories of the physical system.

Similarly, the Farmbot is a CNC robot capable of growing various species of leaf and fruit crops in a hydroponic drip irrigation system. Their creators provide users with the necessary knowledge and skills to download the CAD models and programming codes; however, users who do not have technical knowledge can order their Farmbot ready for assembling [52].

An interesting concept for the development of technology dedicated to small scale farming has been introduced by Mauricio-Moreno et al. [53], through a production model known as S$^3$ that encompasses three elements:

- Sensing. The capacity of a system for event detection, data acquisition, and accurate measurement of changes in the physical parameters of the environment.
- Smart. The capacity of a system to incorporate control and actuation functions that, after the interpretation of input data, supports the decision-making process, following predictive or adaptive logics. Furthermore, the term smart adduces to the ability of several interconnected systems to operate simultaneously.
- Sustainable. This concept applies to the development of technology with a combined social, economic, and environmental perspective.

Miranda et al. [54] applied the S$^3$ concept to develop new technologies that respond to current needs of agri-food industries; in particular, the development of an intelligent hydroponic greenhouse for tomato production. By developing custom technology based on microprocessors and data acquisition boards, they saved on technology investment.

## 11. Small and Medium-Scale Hydroponic Production Systems: How to Select the Appropriate Level of Technology?

As mentioned above, population density is concentrated mainly in urban areas; therefore, in this section, we propose two options to select the appropriate technological environment that best suits the needs of small and medium farmers in these environments. We suggest starting by defining the type of need to satisfy: do you want to acquire a food production system for self-consumption purposes or entrepreneurial purposes? The answer to this will trigger the different technological proposals. The technology level selection process for the implementation of a hydroponic food production system in urban or peri-urban environments is shown in Figure 5.

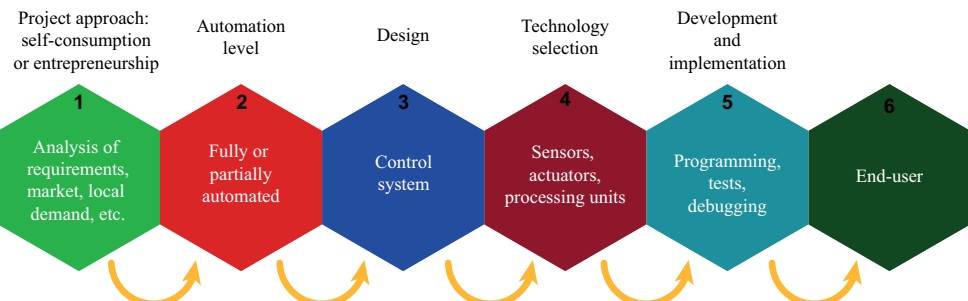

**Figure 5.** Steps for the integration of technology in small-scale food production systems based on the farmer's needs.

For example, suppose the end-user is a person who intends to produce their food. In that case, we recommend implementing a semi-automated system, which will provide a balance between the cost of the system and the long-term benefit. Continuing with this example, the user will need to have adequate space for the design, for example, installing a small greenhouse in their backyard or on the roof of their house. Said space must cover natural light requirements of between 8 and 12 h and adequate natural ventilation to guarantee a constant replenishment of oxygen inside the greenhouse. In terms of automation, pH and electrical conductivity sensors can be installed that notify the user through a cell phone application when it will be necessary to replenish nutrients or adjust pH, this every certain number of days. Irrigation can be easily automated in this scheme using a timer programmed on an Arduino or Raspberry Pi platform.

The second approach corresponds to an automated hydroponic system for medium-scale food production for entrepreneurship purposes. From this perspective, the end-user needs an appropriate initial investment to choose suitable sensors, including actuators and processing elements. We must consider climatic conditions to determine which sensors and actuators are necessary to control variables such as temperature and humidity. In this framework, different data acquisition and processing architectures store the measurements of the variables, among which the Arduino and Raspberry Pi boards stand out for their easy implementation and affordable price. Different control algorithms are programmed within these signal processing architectures to maintain the measured variables in precise crop ranges, achieved by implementing control techniques such as fuzzy logic, PID controls, and even algorithms based on neural networks for image analysis. Finally, based on the control laws, signals are sent to the actuators to manipulate pumps, valves, heating, and cooling systems, among others, to close the control loop and maintain the variables in the desired ranges. Users can monitor the system's status and interact with it through local human-machine interfaces (HMI), phone applications, or websites, giving the user the ability to make timely decisions in system failures or deviations. All the above can coexist in a small and medium scale production environment, for example, vegetable production in small greenhouses located on rooftops or production systems in closed rooms, using artificial light.

Table 6 summarizes some technological applications for small-scale hydroponic systems, emphasizing whether such systems include an Industrial (I), Domestic-Urban (DU), Sustainable (S) and Smart (SM) approach. The criteria to decide whether an application is sustainable or not, depend on the management of variables for saving specific resources, including water, electricity, labor, or economic resources.

**Table 6.** Technological contributions to the automation of tasks in small hydroponic greenhouses.

| ID | Year | Contribution/Author | I | DU | IoT | S | SM |
|----|------|---------------------|---|-----|-----|---|-----|
| 1 | 2021 | Design and implementation of an intelligent, low-cost IoT-based control and monitoring system for hydroponics greenhouses [55] | X | | X | | X |
| 2 | 2021 | IoT-based self-adaptive hydroponics care system that controls the hydroponic cultivation environment [56] | X | | X | | X |
| 3 | 2019 | Development of a methodology for the implementation of sustainable technology [54] | | X | X | X | X |
| 4 | 2019 | Development of an IoT application for task management in a hydroponic greenhouse [57]. | | X | X | | X |
| 5 | 2018 | Integrated Internet of Things (IoT)-based system for monitoring and managing a hydroponic crop [58]. | | X | X | X | X |
| 6 | 2018 | Development of an automated system for nutrient dosing, pH regulation, conductivity, light intensity, and temperature and humidity monitoring using an ARM Cortex-M4 microcontroller [59]. | | X | | | X |
| 7 | 2018 | Development of a microelectronic system for mixing nutrient solutions in hydroponic crops using fuzzy logic [60]. | | X | | | X |
| 8 | 2018 | Modeling of a prototype greenhouse for the control of temperature, humidity, luminosity and nutrient management [61]. | | X | | | X |
| 9 | 2018 | Conductivity adjustment of the nutrient solution in an NFT type hydroponic system using fuzzy logic [62]. | | X | | | X |
| 10 | 2018 | Development of an intelligent hydroponic system based on IoT using deep neural networks in a Raspberry Pi3 and Tensor Flow [63]. | X | X | X | | X |
| 11 | 2018 | Development of a system of monitoring and control of variables for hydroponic crops based on the Internet of Things [64]. | | X | X | | X |
| 12 | 2018 | Development of a nutrient flow control tool in a hydroponic system using Arduino, with remote monitoring and operation using a smartphone [65]. | | X | X | | X |
| 13 | 2018 | Development of an automated data acquisition system for monitoring and control of a hydroponic crop implemented in the Arduino platform [66]. | | X | | | X |
| 14 | 2017 | Low cost control system for hydroponic greenhouses using an AVR microcontroller and an interface developed in LabView [67]. | | X | | | X |
| 15 | 2017 | Development of an intelligent hydroponic system using exact inference through a Bayes-type neural network [68]. | | X | | | X |
| 16 | 2017 | Design and implementation of an automatic system for the dosage of nutrients in an NFT type hydroponic system using Arduino [69]. | | X | | | X |
| 17 | 2017 | Conductivity and pH adjustment of the nutrient solution used in hydroponic cultures using a linear regression model programmed in a microcontroller [70]. | | X | | | X |
| 18 | 2017 | Design of an embedded system for the dosage of nutrients in a hydroponic system using artificial intelligence [71]. | | X | | | X |
| 19 | 2017 | Development of a remote control system based on IoT for the management of a hydroponic system [72]. | | X | | | X |

## 12. Conclusions

This manuscript describes the theoretical and technological aspects of hydroponic-based food production for its implementation on a small and medium scale, providing guidelines for those who wish to contribute to solving the global demand for food that is coming sustainably and efficiently for the year 2050.

Prospects for the market's growth in food production using hydroponics maintain an upward trend for the next 20 years; however, a decentralized production approach is necessary, where small producers in densely populated areas reduce the ecological footprint derived from this activity.

Although the initial investment is usually high, the implementation of technology in small and medium-scale decentralized food production systems can positively impact local economies by promoting self-employment or profitable business activities and favoring an environment of cooperation in the communities.

Platforms such as Raspberry Pi and Arduino, together with a new range of sensors and actuators currently available on the market, will revolutionize small and medium-scale precision agriculture, allowing the optimization of production in more controlled environments at affordable prices for the farmer.

**Author Contributions:** Conceptualization, R.S.V.-G., J.C.S.-S. and A.L.G.-G.; investigation, R.S.V.-G. and A.L.G.-G.; review of technical concepts, A.L.G.-G., J.D.O.B.-S. and E.V.-Z.; writing—review and editing, R.S.V.-G., J.C.S.-S. and A.L.G.-G.; funding acquisition, J.C.S.-S. All authors have read and agreed to the published version of the manuscript.

**Funding:** This work was partially funded by SIP-IPN 20220086 for Julio Sosa.

**Institutional Review Board Statement:** Not applicable.

**Informed Consent Statement:** Not applicable.

**Data Availability Statement:** Not applicable.

**Acknowledgments:** This work was supported by Instituto Politécnico Nacional, trhough grant SIP 20220086. Roberto Velazquez acknowledges financial support from Consejo Nacional de Ciencia y Tecnologia (CONACyT) to carry on graduate studies.

**Conflicts of Interest:** The authors declare no conflict of interest. The funders had no role in the collection, analyses, interpretation of data, writing, or deciding to publish this work.

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
