# Peer review of "A Review on Hydroponics and the Technologies Associated for Medium- and Small-Scale Operations"

_agriculture, doi:10.3390/agriculture12050646_

Round 1

Reviewer 1 Report

Dear appreciated authors,

The manuscript presents quality and interesting paper. Several details are remarked in revised pdf file, in comments for each row/line.

The details which could be corrected are:

Line 116: Instead whole sentence "Hydroponics, derived from the Greek h´ydor = water and ponos = work, is a type of horticulture and a subset of hydroculture, a method that uses nutrient mineral solutions 
instead of tillage [13]",
it could be enought:
Hydroponics present a type of horticulture and a subset of hydroculture, a method that uses nutrient mineral solutions instead of tillage [13]-

Line 192: There is one more space between point and  last word in sentence (crews . Additionally)

Line 504: The sentence "This manuscript describes the theoretical and technological aspects of hydroponic based food production for its implementation on a small and medium scale, providing guidelines for those who wish to contribute to solving the global demand for food that is coming sustainably and efficiently for the year 2050." could be placed behind Conclusion , as a first sentence (in line 492). 

Reviewer 2 Report

Review

A review on hydroponics and the technologies associated for medium- and small-scale operations

In general, the submitted MS is interesting and will call the attention of specialists; only English style and grammar need to be verified.

Observations and suggestions

Page 1

It says. - Lines 6-7:

Although hydroponics has proven its effectiveness on a large scale, there are still challenges 6 in implementing this technique on a small scale; specifically in urban settings.

Suggestion:

Although hydroponics has proven its effectiveness on a large scale, there are still challenges in implementing this technique on a small scale, specifically in urban and suburban settings. Also, in rural communities, the availability of technological devices and appliances is frequently scarce.

Page 2

It says. – Lines 47-49:

Regions where arable land and water is becoming scarce, are turning to high-yielding farming methods and technologies, like hydroponics, to meet their demand for healthy, affordable, and sustainable food.

Suggestion:

Those productive regions, where arable land and water are becoming scarce, are turning to intensive high-yielding farming methods and technologies, like hydroponics, to meet their demand for healthy, affordable, and sustainable food.

Page 3

It says. – Lines 123-128:

…. Located in que outskirts of south Mexico City, Xochimilco’s one thousand-year-old network of canals 124 and artificial islands lionize the ingenuity of the Aztec people to make a sustainable habitat out of a wetland. Additionally, by being recognized by UNESCO as a World Heritage Site, it epitomizes the potential of UA not only as a sustainable food source, but also as an influential element in both social and environmental welfare.

Suggestion:

Located in the outskirts of south Mexico City, Xochimilco’s one thousand-year-old network of canals 124 and artificial islands lionize the ingenuity of the Aztec people to make a sustainable habitat out of a wetland. In addition, it epitomizes the potential of UA as a sustainable food source and as an influential element in both social and environmental welfare. For the above, UNESCO recognized this hydro-system as a World Heritage Site.

Page 4

It says. – Lines 145-150

Hydroponics, unlike traditional farming, does not require soil to grow food. In general, this technique consists of cultivating plants on natural or man-made substrates, where nutrients are absorbed from a nutrient solution through the plant’s roots. There are different methods for growing food using hydroponics and their application depends on the specificc plant, local climate, and budget, among other factors. Most systems comprise a storage 149 tank for the nutrient solution and an aerator, as illustrated in Figure 1.

Suggestion:

Hydroponics, unlike traditional farming, does not require soil to grow food. In this technique, plants are grown, either on a natural or on a particular 'man-made' substrate, where the roots easily extract the nutrients from a prepared nutrient solution. There are different methods for growing food using hydroponics, and their application depends on the specific plant, local climate, and budget, among other factors. Most systems comprise a storage tank for the nutrient solution and an aerator, as illustrated in Figure 1.

Page 5

It says. – Lines: 207-209

  • * Operations in hydroponics are much simpler than in traditional agriculture, which requires many effort-laden preparations before sowing, including the cost of heavy machinery and specialized equipment, even if they are only rented.

Suggestion:

  • In some routinary steps, operations in hydroponics are simpler than those required in traditional agriculture. In this sense, conventional practices require many effort-laden preparations before sowing, including the cost of heavy machinery and specialized equipment, which eventually may come through a rental of the same. In other aspects, hydroponics may require more dedication, commonly needing a set of sensors and devices for a precise follow-up of the crop condition.

Page 6

It says. – Lines 232-238

  • Chemically inert. The substrate must be made of materials that do not react with the chemicals in the nutrient solution to avoid any alteration in its composition.
  • Biologically inert. Because the nutrient solution circulates among several plants, diseases can spread rapidly from one to another if corrective actions are not taken immediately; therefore, the substrate must not favor any biological activity, since micro-organisms may have a detrimental effect on crops, like diseases, malnutrition, etc.

Suggestion:

  • Chemically inert. The substrate must consist of materials unable to react with the chemicals in the nutrient solution to avoid any alteration in its composition.
  • Biologically inert. Because the nutrient solution circulates among a high root density of several plants, diseases can spread rapidly from one to another if corrective actions are not applied immediately. Therefore, the substrate must not favor any biological activity since micro-organisms may have a detrimental effect on crops, like diseases, malnutrition, and other consequences.

Page 7

It says. – Table 1. Cont.

Left column:

Peat

Suggestion:

Peat/Peat moss

Page 8

It says. – Lines: 273-274; 276-277

…. Finally, during fruit ripening, the plant requires nutrient solutions low in nitrogen and high in potassium concentrations.

… For instance, an 8-15-36 formulation, ideal for tomato crops, contains 8% nitrogen, 15% phosphorous,

Suggestion:

…. Finally, during fruit ripening, the plant requires nutrient solutions low in N and high in K concentrations.

… For instance, an 8-15-36 formulation, ideal for tomato crops, contains 8% N, 15% P,

Page 9

It says. – Line: 278

…. and 36% potassium.

Suggestion:

…. and 36% K.

Page 10

It says. – Lines: 284-287

To exclude changes in the nutrient solution, the volume in the solution storage tank must be constant, replenishing the water absorbed by the plants or evaporated; otherwise, the concentration of the salts will change, affecting the healthy growth of the plants.

Suggestion:

To exclude changes in the nutrient solution, the volume in the solution storage tank must be constant, replenishing the water absorbed by the plants and lost by evapotranspiration; otherwise, the concentration of the salts will change, affecting the healthy growth of the plants.

Page 11

It says. – Lines: 311-317

An aseptic environment is of the utmost importance in hydroponic systems to efficiently produce good quality produce; however, in practice, it is difficult to keep completely sterile the zone around the plant’s roots [35]. The most obvious symptom of disease in a plant is leaf withering caused by Fusarium and Verticillium fungi. Other species of parasites like Pythium and Phytophthora also represent menaces to the plant’s roots. Unfortunately, there are no fungicides safe enough to use in hydroponics without compromising consumer health [36].

Suggestion:

An aseptic environment is of the utmost importance in hydroponic systems to efficiently produce good quality products; however, it is hard to keep full sterile the zone around the plant’s roots [35]. The most obvious symptom of a disease in a plant is leaf withering caused by Fusarium and Verticillium fungi. Other species of parasites like Pythium and Phytophthora also represent menaces to the plant’s roots. Unfortunately, there is no safe enough fungicide to use in hydroponics without compromising consumer health [36].

Page 12

It says. – Lines: 331-333

In open-sky cultivation, rain may carry excess nutrients to rivers, eventually reaching lakes and oceans. For instance, recent studies have determined that the so-called death zone in the Gulf of Mexico (Figure 3), occurring annually, is due to the discharges …

Suggestion:

In open-field cultivation, rain may carry an excess of nutrients to rivers, eventually reaching lakes and oceans. For instance, recent studies have determined that the so-called death zone in the Gulf of Mexico (Figure 3), occurring annually, is due to the discharges …

Page 13

It says. – Lines: 335-341

…….. Once the polluted waters reach the ocean, the excess of nutrients encourages the inordinate growth of algae that, after dying, sink and decay, consuming high levels of oxygen. Without enough oxygen, marine life is under stress, with long-term impact on species that cannot leave the area [40]. On the other hand, a sustainable way to manage residual nutrient solutions in hydroponics is achieved by implementing closed systems that recover the water at the end of the cycle through treatment and sterilization, as schematized in Figure 4

Suggestion:

Once the polluted waters reach the ocean, the excess of nutrients encourages the extraordinary growth of algae that, after dying, sink and decay, consuming high oxygen levels. Without enough oxygen, marine life is under stress, with a long-term impact on species that cannot leave the area [40]. On the other hand, a sustainable way of managing residual-nutrient solutions in hydroponics could be by implementing closed systems that recover the water at the end of the cycle through treatment and sterilization, as schematized in Figure 4

Page 14

It says. – Lines: 370-375

Hydroponics fits perfectly into the frame of Agriculture 4.0, as large companies make increasing use of breakthroughs in indoor vertical farming, artificial intelligence, and plant biology, to grow an extensive line of products [43]. With the support of advanced and disruptive technologies, and sound scientific knowledge to warrant high yield, we can safely say that hydroponics has clenched a center place in the food production systems of the future.

Suggestion:

Hydroponics fits perfectly within the frame of Agriculture 4.0, as large companies are increasing the use of breakthroughs in indoor vertical farming, artificial intelligence, and Plant Biology to grow an extensive line of products [43]. With the support of advanced and disruptive technologies and the sound of scientific knowledge to warrant high yield, we can say, with certainty, that hydroponics has clinched a center place in future food production systems. 

Page 15

It says. – Lines: 402-409

To contribute significantly to reach SDG 11, such technologies must be scalable to fulfill the needs of growers, not only for large-scale operations, but also for medium- and small-scale, considering the limitations of space available for farming. Buehler and Junge [49] have shown that the practice of growing food in urban environments has grown significantly since 1988 in Europe, Asia, and North America, where the dedicated surface is larger than 150,000 m2. The practice of hydroponics is key for UA, and it may contribute significantly to reach SDG 11; however, it requires the development and adoption of appropriate technologies.

Suggestion:

For a significant contribution to reaching SDG 11, such technologies must be scalable to fulfill the needs of growers, not only for large-scale operations but also for medium- and small-scale, considering the limitations of space available for farming. 404 Buehler and Junge [49] have shown that the practice of growing food in urban environments has grown significantly since 1988 in Europe, Asia, and North America, where the dedicated surface is more than 150,000 m2. The practice of hydroponics is a key for UA, and it may contribute significantly to reaching SDG 11; however, it requires the development and adoption of appropriate technologies.

Page 16

It says. – Lines: 467-477

The second approach corresponds to an automated hydroponic system for medium-scale food production for entrepreneurship purposes. The end-user will need an adequate initial investment to acquire the appropriate sensors, actuators, and processing elements from this perspective. We must consider climatic conditions to determine which sensors and actuators are necessary to control variables such as temperature and humidity. The measurements of the variables are sent to different data acquisition and processing architectures, among which the Arduino and Raspberry Pi boards stand out for their easy implementation and affordable price. Different control algorithms are programmed within these signal processing architectures to maintain the measured variables in appropriate ranges for the crops; this is achieved by implementing control techniques such as fuzzy logic, PID controls, and even algorithms based on neural networks for image analysis.

Suggestion:

The second approach corresponds to an automated hydroponic system for medium-scale food production for entrepreneurship purposes. From this perspective, the end-user needs an appropriate initial investment to choose suitable sensors, including actuators and processing elements. We must consider climatic conditions to determine which sensors and actuators are necessary to control variables such as temperature and humidity. In this framework, different data acquisition and processing architectures store the measurements of the variables, among which the Arduino and Raspberry Pi boards stand out for their easy implementation and affordable price. Different control algorithms are programmed within these signal processing architectures to maintain the measured variables in precise crop ranges, achieved by implementing control techniques such as fuzzy logic, PID controls, and even algorithms based on neural networks for image analysis.

Page 17

It says. – Lines: 488-490

The criteria used to decide whether an application is sustainable or not, is based on the management of variables to save specific resources such as water, electricity, labor, or economic resources.

Suggestion:

The criteria to decide whether an application is sustainable or not depend on the management of variables for saving specific resources, including water, electricity, labor, or economic resources.

Reviewer 3 Report

Below are some suggestions and comments for the authors to consider:

Table 2: Consider adding suggested cultivation techniques as a separate column to this table as that will be helpful to readers.

Table 3: Double check this table and correct all inaccuracies. The symbols for phosphorus and potassium are switched and so are their absorbable forms.  More so, the absorbable forms of some of the nutrients listed are inaccurate. Below are the suggestions for correction:

Nutrient

Symbol

Forms Absorbed

Nitrogen

N

NO32-, NH4+

Phosphorus

P

PO43-, HPO42-, H2PO4-

Potassium

K

K+

Sulfur

S

SO42-

Molybdenum

Mo

MoO42-

Boron

B

BO32-, B4O72-

Line 279: It is an unclear how compost which is solid can be added into a hydroponic system which is mainly water with nutrients running through pipes.  Are the authors referring to compost tea?  This needs clarification.

Figure 2: At high pH conditions, calcium tends to bind to phosphorus to form calcium phosphate making it less available to plants so please depict this in your figure.  There is a way to illustrate this.

Line 299: Insert the correct figure number instead of figure??

Line 325: Consider using open field instead of open sky.

Line 32: Replace death zone with dead zone.

Lines 446 -490: Adding cost analysis for small and medium scale hydroponic systems that addresses their potential profitability to Section 11 will enrich the manuscript.
